# Research on Spatial Planning of Petrochemical Industrial Parks from the Perspective of Symbiosis: Example of Yueyang Green Chemical Industry Park

**Min Wang [1,2,\*], Xiaohan Yuan [2], Shuqi Yang [2], Kahaer Abudu [2] and Kongtao Qin [2]**

[1]   Key Laboratory for Geographical Process Analysis & Simulation Hubei Province, Central China Normal University, Wuhan 430079, China
[2]   College of Urban and Environment Science, Central China Normal University, Wuhan 430079, China; yuanxiaohan123021@163.com (X.Y.); 13643438802@163.com (S.Y.); 18971099080@163.com (K.A.); qkt777@163.com (K.Q.)
\*   Correspondence: campero@ccnu.edu.cn; Tel.: +86-027-6786-8305

**Abstract:** As a practical exploration of industry ecologicalization, ecoindustrial parks (EIP) serve as an effective approach to sustainable development. Different from western industrialized countries, China is accelerating its industrialization, and the philosophy of symbiosis is embodied more in the requirements of economy and environmental protection in the production process than in the long-term social and environmental mutual construction. EIPs are a regional system consisting of nature, industry, and society, and the key to achieve industrial symbiosis is systematically allocating resources for industry, city, and people through planning. Based on engineering practice, the authors selected the special space of China Yueyang Petrochemical Industrial Park as the object of discussion, addressing the main problems and challenges facing its development, and focusing on the relationship between industrial symbiosis and spatial symbiosis. From the analysis of the current situation, the circular symbiotic industrial chain network, and the layout of the symbiotic park are included in the spatial planning of the industrial park.

**Keywords:** symbiosis; petrochemical industry; planning; Yueyang Green Chemical Industrial Park

## 1. Introduction

An industrial park is not only a spatial carrier for industrial production, but an important organizational model for industrial development. Petrochemical parks have become the focus of environmental and economic conflicts due to their large scale of industrial activities, strong intensity, density of material energy, and high risks of environmental safety. For this reason, recycling transformation of parks, ecoindustrial parks, and green park planning have emerged globally. In China, with the in-depth study of industrial symbiosis and industrial ecosystems, the planning and construction of industrial parks under the symbiosis concept have been carried out. Industrial symbiosis (IS) refers to the cooperation between different enterprises, using this cooperation to jointly improve the viability and profitability of enterprises, and ultimately achieve resource conservation and environmental protection [1]. The core of IS is to divide the enterprises in the park into producers, consumers, and decomposers, and to form an enterprise alliance with circular symbiosis characteristics through resource exchange [2,3]. At present, domestic and foreign scholars have conducted research on the enterprise chain network [4], industrial efficiency [5], clean production, and park management [6] of industrial parks from the perspectives of industrial science, economics, ecology, and management, and proposed core paradigms: end-of-pipe management with waste treatment [7], clean production with production process improvement [8], lifecycle management with product improvement [9], or green development with overall system optimization [10]. These paradigms mainly work

at the enterprise and industry chain levels, leading to the current common implementation of ecoindustrial parks (EIP) and green park transformation. However, the construction of these industrial cycle systems does not cover the specific requirements of spatial implementation, and it is not easy to implement them into the spatial design of symbiotic industrial parks. Urban and rural planning scholars start from the organization of material space such as site layout, park design, and landscape system to propose implementation plans. Since the spatial design is based on traditional industrial building codes, the special characteristics of different industrial sectors and economic bases are less considered. In particular, the construction of most industrial parks faces the upgrading and transformation from the traditional economic model to the green symbiosis model, and there are difficulties in the interface between their industrial planning and spatial planning in terms of preparation timing, planning content, and implementation mechanism. Therefore, to address the problem of mutual disconnection between the industrial chain network and land-use planning in China's symbiotic industrial park planning, this study attempts to consider the industrial layout and land-use planning together, to continue the technical route of the current situation analysis of the park, construction of industrial chain network, evaluation of environmental bearing, and generation of the land-use plan by combining qualitative and quantitative methods, and to apply Yueyang Yunxi Petrochemical Industrial Park as empirical evidence to explore the method of land-use planning in a symbiotic industrial park.

Industrial land is a key space for the development and optimal utilization of national land space, which is of great significance in the process of urbanization and sustainable development. The current spatial planning of industrial land focuses on land selection and landscape planning [11], and introduces quantitative methods such as ecologically sensitive land division [12] and environmental carrying capacity evaluation [13], which provide a certain theoretical basis for the scientific land use of industrial parks. In practice, in order to maximize the spatial benefits of land, planning must give attention to the current situation of the industrial park to reduce the disturbance and impact on the surrounding urban environment. A comprehensive spatial planning strategy that integrates land-use layout, ecological impact, spatial form, and social effects is formulated [14]. Therefore, based on the analysis of the current situation, it is the focus of this paper to propose design strategies and methods to adapt to the current environment. The difference with traditional industrial parks is that symbiotic industrial parks not only emphasize the maximization of economic profit, but give more attention to the coordinated development of economic, environmental, and social functions [15]. Thus, the spatial planning of an industrial park under symbiosis concept introduces the concept of coordinated development of industry, city, and people. Initially, the industrial symbiosis chain network is the basis of spatial planning, promoting production with cities, and solidly promoting industrial development in urban planning. Specifically, it finds the functional positioning of the park in the context of symbiosis, clarifies industrial development goals, and guides the park and the city to depart from the traditional development framework, forming a stepwise, progressive development trend. The spatial planning of the park is based on the objective demand of the industrial development chain, and the development space is reserved for the advantageous industries. Secondly, the sustainable development of industry is closely related to the quality of the ecological environment. Based on the carrying capacity of the ecological environment, the living environment of residents is optimized through orderly construction, and the virtuous cycle of life, production, and ecosystem is promoted. Finally, improve infrastructure and strengthen industrial advantages. Systematic and forward-looking planning of infrastructure enables lasting support for industrial development. Based on these features, this paper follows the vein of current situation analysis, industrial research, ecological carrying capacity evaluation, and infrastructure construction to explore the path of spatial planning practice for a petrochemical industrial park from the perspective of symbiosis.

Hunan Yueyang Green Chemical Industry Park (formerly Yunxi Industrial Park) is a provincial-level economic and technological development zone approved by Hunan Provincial People's Government in August 2003. Relying closely on the resource advantages of Baling Petrochemical and Changling Refinery in the park, the park has been vigorously developing fine chemicals in accordance with the idea of "establishing the park with special features and developing the park with science and technology" and the purpose of "docking the petrochemical base, undertaking coastal industries, and creating an industrial depression". In response to the requirements of healthy, green and sustainable development, the park has gone through three rounds of planning to achieve the strategic transformation from a healthy, clean, green and ecological park to a symbiotic park. The current construction of the park has the following problems: (1) Environment—Environmental pollution seriously affects the sustainable development of the park. (2) Power—The depth of product processing is insufficient, and the industrial chain needs to be further extended. At present, some chemical enterprises only process primary raw material resources, and the added value of products is low. Although the park's chemical industry has taken shape, the enterprises still need to strengthen the scale effect. (3) Transformation—Yueyang Green Chemical Industry Park is in the middle and late stage of industrialization, which is both energy consuming and emitting, and is the leading industry and the foundation of Yueyang. With the green transformation and development, recycling and symbiosis in the industrial park is imminent. (4) Planning—The previous rounds of industrial park planning focused on industrial symbiosis, the introduction of new industries, and the optimization and upgrading of the industrial chain. The concept of industrial symbiosis has not yet been integrated into the spatial construction, as the comprehensive consideration of industry, city, and people in spatial planning is insufficient. This study aims to explore the symbiosis concept in the industrial design and park layout of petrochemical industrial parks, and further extends it to the synergistic development of industry, city, and people in symbiotic cities, which will enrich the practical exploration of symbiotic industrial parks and symbiotic small towns.

## 2. Theoretical Background

### 2.1. Industrial Symbiosis Theory

Ayres, a pioneer in the field of industrial metabolism, has made useful explorations in the nature, behavioral mechanisms, and application values of industrial symbiosis in the context of industrial ecology, proposing that industrial symbiosis is a model of industrial production by simulating natural ecosystems [16]. In analogy with the symbiotic mechanism of natural ecosystems, industrial symbiosis is the result of equipment sharing, waste-stream concentration, and energy exchange among enterprises [17], in which the driving mechanism is more of a cooperative symbiotic relationship established by the pursuit of common interests [18]. The agglomerative development of enterprises in industrial parks reflects a more comprehensive competition and cooperation between ecosystems and industrial systems [19]. Therefore, government management and laws and regulations can accelerate industrial symbiosis development [20], and spatial commercial proximity becomes a determinant of industrial symbiosis [21]. When planning symbiotic industrial parks, we focus on the design, construction, and maintenance research of symbiotic industrial chains based on industrial symbiosis theory, and propose four types of symbiotic networks: dependent, equal, nested, and virtual [22]. The inner connection of symbiotic unit, symbiotic mode, and symbiotic environment is considered in the industrial symbiosis mechanism, and the symbiosis mechanism is promoted to function through policies, regulations, and institutions [23].

### 2.2. Symbiotic Industrial Park

Symbiotic industrial park is a practical form of industrial sustainable development, which forms an industrial symbiosis system in terms of elements through the sharing of enterprises, energy, and infrastructure, improving the ecological efficiency and com-

prehensive competitiveness of the park. Foreign industrial parks with typical symbiotic characteristics include Kalundborg Industrial Park in Denmark [24], Choctaw Industrial Park in the United States [25], Ulsan Ecological Industrial Park in South Korea [26], and Kawasaki Industrial Park in Japan [27]. They have flexibly applied the theories of cleaner production, circular economy, and industrial symbiosis to construct a circular symbiotic industrial chain network and achieve the goal of efficient utilization of materials and energy. China started the pilot construction of national ecoindustrial parks in 1999, from the first national ecoindustrial park in Guangxi, Guigang Eco-Industrial Park [28], to the construction of Jiangsu Economic Development Zone, which has explored from multiple perspectives the concepts of industrial systems, management methods, and benefit evaluation. In the construction of Guigang Industrial Park in Guangxi, the overall planning and design of the park is coordinated with the extension of the industrial chain, and the production technology, industry, and park energy, and products are considered in an integrated manner, which provides useful reference for the construction of agro-industrial symbiosis-type industrial parks in the western region.

Essentially, the development of symbiotic industry is the process of realizing park ecology, and its ideological origin starts from the observation of industrial symbiosis phenomenon and the analogy between industrial ecosystems and natural ecosystems. The two most important parts of park construction are industrial planning and spatial planning. In terms of the relationship between the two planning parts, industrial planning is the basis of spatial planning; industrial planning is the basis of spatial planning, which determines the industrial type, industrial chain relationship, and production process of the park. Spatial planning is the material realization of industrial planning, coupled with the industrial organization mode and circular chain network. The spatial scale, land type, and facility support match with industrial symbiosis system to realize material exchange and energy level gradient utilization in production and consumption process. At the spatial level, it is especially necessary to analyze the reciprocal feedbacks between the industrial, infrastructure, and ecological elements of the park, and to emphasize the evaluation of the adaptability of artificial and ecological elements.

### 2.3. Theory of Ecological Carrying Capacity

Ecological carrying capacity theory states that accurate calculation of the carrying capacity of the Earth and the resilience of the Earth to human economic activities is a fundamental issue for sustainable development strategies [29]. Ecological carrying capacity includes the self-regulating capacity of ecosystems, the ability to recycle resources, and socioeconomic pressures. The structure, processes, and spatial patterns of ecosystems in a particular region determine the population and economic size supported by ecosystem services. The inherent natural attributes of the area determine land use and its suitability. Reasonable use of environmental capacity realizes the symbiotic process of human and natural environment, regional development, and the sustainable development of regional economy, society, and ecology.

### 2.4. Theory of Sustainable Development

The essence of sustainable development is to achieve a dynamic balance between economic development and environmental ecology. Sustainable development is achieved by controlling population expansion, protecting resources, and developing efficient, green, and renewable energy sources [30]. Its ecological significance is to follow the development of the regenerative capacity of the environmental system, so that the living environment of humans can be sustained. Its social significance lies in creating a good living environment for humans [31]. It can be said that sustainable development is the joint development of human society and ecological environment. Based on this, focusing attention on the ecological rationality of economic activities, supporting economic activities that are beneficial to resources and the environment, and reducing economic activities that waste resources

and damage the natural environment similarly constitute the basic principles of symbiotic industrial park planning.

In summary, this paper focuses on the ecological–land-use coordination relationship in the spatial planning of petrochemical industrial parks under the concept of symbiosis. Furthermore, we explore the land-use planning method of ecological symbiosis in petrochemical industrial parks by introducing the ecological carrying capacity model and the prediction model of industrial land use under ecological constraints, combining qualitative and quantitative industrial symbiosis demand and land-use capacity prediction on the basis of the petrochemical industry symbiosis chain. In theory, we try to integrate urban and rural planning methods into industrial ecology, and, in practice, explore the spatial layout mode of petrochemical industrial parks with the goal of ecological symbiosis in practice.

## 3. Materials and Methods

### 3.1. Studied District

The city of Yueyang is an emerging suburb industrial district featuring dual development of industry and agriculture and the combination of urban and rural areas. Yueyang Green Chemical Industrial Park possesses an entire industrial chain from petroleum catalyzing to petroleum cracking to petrochemicals. Relying on the former Yunxi Industrial Park, with the two major plants of Baling Petrochemical and Changling Refinery as the leaders, the "one park and three pieces" pattern is formed. The "one park" is Yueyang Green Chemical Industry Park, and the "three pieces" are Yunxi City area (Baling Petrochemical Plant, Yunxi Fine Chemical Park, and New Material Industry area), Changling area (Changling Refinery Plant and Changling Industrial Park), and Linxiang Binjiang Industrial Park in the Baling area. This paper focuses on exploring the symbiotic planning and design of the industrial park in the Changling area under the requirement of technological reform and upgrading. The park administers two towns and one subdistrict, namely, Lucheng Town, Lukou Town, and Changling Subdistrict, and there are large state-owned enterprises of Sinopec Changling Refining and Chemical (Sinopec Group Changling Refining and Chemical Co., Ltd., and China Petroleum and Chemical Corporation Changling Branch) within the planning area of the park [32]. The main problem facing the Changling area is that the production capacity is insufficient, the industrial chain is not perfect, the direction of spatial expansion is unknown, and it is necessary to accelerate the development of the integration of industry and city. Compared with other well-developed petrochemical parks, the current production capacity of 2 million tons per year is still small and medium-sized refineries, and its supporting chemical production units are mainly below 100,000 tons, and the main products of processing enterprises are refined oil, and less than 10% of chemical raw materials such as ethylene, aromatic hydrocarbons and olefins, which have high added value and can drive the sustainable development of downstream industries, are available. The buildable land in the park is basically exhausted, and the usable area cannot be expanded [33].

Based on this, along with several technical transformation projects in the original old factory area to be moved out, in order to cooperate with the "technical transformation and expansion" plan of the plant area, a new batch of crude oil processing and fine chemical projects have been added, and the planning of the petrochemical industrial park in Changling area has been considered from the aspects of project placement, enterprise configuration, space site selection, development timing, etc. A few crucial questions need to be answered: Is it possible to continue to expand the construction of industrial parks in the Changling area? What is the direction of industrial development in the Changling area? What should be the proper size of the district's industrial development? Where is the space for industrial development? Which industry agrees with the district? How will industrial development pose impact on the environment? What is the best strategy for site selection and layout pattern for the Changling District of Yueyang Green Chemical Industrial Park? Only when the strategic problems are solved can Changling Industrial District make progress in a correct direction, and its space and facilities safeguarded [34].

### *3.2. Data Source and Processing*

### 3.2.1. Source of Data

The basic information used in this study includes topographic maps of the study area, photos of the current situation, socioeconomic statistics of the town, industrial development information of the industrial park, and relevant planning information. The basic data are mainly obtained from field surveys and base research, as well as statistical analysis after obtaining initial information by means of departmental interviews. The topographic maps, laws, and regulations required for the planning are from the Bureau of Land and Resources, including the Urban and Rural Planning Law of the People's Republic of China, Measures for the Preparation of Urban Planning, Urban Land Use Classification and Planning and Construction Land Use Indicators, Land Use Control Indicators for Industrial Projects, Yueyang City General Plan (2007–2020), Yueyang Yunxi District Zoning Plan (2005–2020), Outline of Industrial Development Plan for Yunxi Industrial Park in Hunan Province (2008), and Statistical Bulletin of National Economic and Social Development of Yunxi District. The industrial resource information of the case area includes the socioeconomic statistical yearbook of Yunxi District, the socioeconomic indexes of the two towns within the case area, the main enterprises and economic indexes of the two towns, and the main economic indicators of petrochemical enterprises in the long refining area. They are the basic information for the subsequent industrial resource analysis, industrial scale demonstration, and industrial chain design.

### 3.2.2. Determine Planning Principles

After analyzing the master plan of the area studied, the future planning direction of the site was determined: (1) Symbiotic development of production and city—By analyzing the role Changling District plays in the whole industrial network of Yueyang and Yunxi District, the spatial development direction, timing, and scale of industrial development in Changling area were reasonably determined. (2) Spatial symbiotic development—Under the conditions of market economy, the development trend and the possible scale of products needed to be analyzed to provide information on the land needed for construction. At the same time, it was important to consider environmental and spatial capacity of the district to choose what to produce upstream and downstream, and extend the industrial chain to abandon projects that go against environmental protection and sustainable development. (3) Social symbiotic development of town, garden, and people—Integrating industrial parks, land, and people based on small cities and towns helps to develop nearby industries. Relying on large petroleum refining and chemical factories, with industry taking the lead, forming a diverse network consisting of industry, logistics industry, special agriculture, service industry, and tourism, was formed, actually realizing the linkage development of "businesses and localities". Based on the principle of sharing resources and infrastructure, the layout of all infrastructure was rationalized, and the needs for long-term development were also fully considered.

### 3.2.3. Analysis of the Status Quo of the Park

The advantages of the park include: accessible location, good resources, industry with outstanding characteristics, and a solid industrial development foundation. The area is very easy to form an industrial symbiosis network and has natural advantages for developing green and sustainable industries. The disadvantage of the park is that it faces the problem of pollution before treatment. The current planning of Yueyang Green Chemical Industry Park only considers industrial symbiosis and focuses on the introduction of new industries and the optimization and upgrading of industrial chains. However, it does not take into account the comprehensive consideration of production, city, and people, and does not systematically integrate the concept of spatial symbiosis. In the external factors, the development of the park has the opportunities of national policies and independent innovation. At the same time, it also faces many threats.

### 3.2.4. Design of Symbiotic Industrial Chain

Taking into account the superior guidance of the Industrial Development Plan of Yunxi Industrial Park in Yueyang City, Hunan Province, combined with the development of domestic and foreign petrochemical industry with the characteristics of new chemical materials and economic chemicals, and the main industrial sectors are determined as petrochemicals, fine chemicals, and processing and manufacturing, all under the concept of symbiosis. Specific industrial chain comparison design was carried out for each industrial sector, and the configuration of upstream, midstream, and downstream industrial chains were finally determined, and implementation requirements put forward in terms of spatial site selection and infrastructure support. Decision-making tools such as lifecycle analysis [35] and environmental indicator analysis [36] were used to help identify and screen enterprises and achieve waste conversion in the park. In addition, industrial planning in the Changling area focuses on the development of industrial chains and product clusters. The main products were integrated and serialized to form an organic linkage, which facilitated the full utilization and optimal allocation of resources. It was finally determined that the industrial symbiosis network in Yueyang Green Chemical Industry Park could be expressed in the following three types: (1) "configuration type" for the purpose of closed circuit system; (2) "waste exchange type" in which the byproduct trade between enterprises could reduce the waste disposal cost of one enterprise and the raw material purchase cost of another enterprise; and (3) "network type", which connects small- and medium-sized enterprises to improve operational efficiency and reduce waste.

### 3.2.5. Ecological Footprint and Ecological Carrying Capacity Estimation

The ecological footprint is the biologically productive land area required for the services consumed by the population of a given region at a given time, and the ecological carrying capacity is the sum of the ecological services that a given region can provide to humans [37].

The calculation formula follows:

$$EF = N \times ef = N \times \sum_{j=1}^{n}(a_j \times r_j) \tag{1}$$

$$EC = N \times ec = N \times \sum_{j=1}^{n}(a_j \times r_j \times y_j) \tag{2}$$

In the equation, EF represents total ecological footprint; N represents current population; ef stands for regional ecological footprint per capita; $a_j$ is the area of land occupied per capita for biological production of category j; and $r_j$ is equivalent factor [38]. The equivalent factor used in this paper is calculated according to equation $r_j = \frac{P_j}{P}$, where $P_j$ is the average productivity of land j, and p is the average productivity of land j nationwide. Relevant data were collected from the statistical yearbook in previous years. EC represents the total ecological carrying capacity of the district; ec represents ecological carrying capacity per capita of the district; $y_j$ is yield factor, the yield factor used in this paper is calculated according to equation $y_j = p_j^z/p_j$, and $p_j^z$ is the average productivity of land j in region z and $p_j$ is the average productivity of land j nationwide. Relevant data were collected from the statistical yearbook in previous years. Lastly, n represents the number of ecosystem species participating in the assessment, and 4 species are calculated in this paper.

Ecological footprint is the demand of the ecosystem and ecological carrying capacity is the supply of the ecosystem, and their comparison can lead to ecological surplus or ecological deficit. If there is an ecological deficit, it indicates that the human load of the area exceeds its ecological capacity; on the contrary, it is an ecological surplus [39]. Using the equations given above, the calculation results are shown in Table 1.

**Table 1.** Statistical analysis of Yunxi District's ecological model.

| Year | Land Type | | Arable Land | Grassland | Forest Land | Water Area |
|---|---|---|---|---|---|---|
| 2015 | Ecological Footprint Demand per capita | Area per capita (hm²/cap) | 0.0513 | 0.0017 | 0.0778 | 0.0459 |
| | | Equivalent Factor | 4.72 | 1.13 | 0.19 | 3.02 |
| | | Equivalent Area (hm²/cap) | 0.2421 | 0.0019 | 0.0148 | 0.1386 |
| | Ecological Carrying Capacity | Area per capita (hm²/cap) | 0.0451 | 0.0015 | 0.0685 | 0.0404 |
| | | Equivalent Factor | 4.72 | 1.13 | 0.19 | 3.02 |
| | | Yield Factor | 3.4633 | 0.0052 | 2.2694 | 7.1128 |
| | | Equivalent Area (hm²/cap) | 0.738 | 0.0001 | 0.0295 | 0.8676 |
| | Ecological Surplus | 1.2377 | | | | |
| 2016 | Ecological Footprint Demand per capita | Area per capita (hm²/cap) | 0.0507 | 0.0017 | 0.0779 | 0.0454 |
| | | Equivalent Factor | 4.39 | 1.44 | 0.18 | 3.22 |
| | | Equivalent Area (hm²/cap) | 0.2226 | 0.0024 | 0.014 | 0.1462 |
| | Ecological Carrying Capacity | Area per capita (hm²/cap) | 0.0446 | 0.0015 | 0.0686 | 0.034 |
| | | Equivalent Factor | 4.39 | 1.44 | 0.18 | 3.22 |
| | | Yield Factor | 3.2273 | 0.0053 | 2.4752 | 7.7436 |
| | | Equivalent Area (hm²/cap) | 0.6321 | 0.0001 | 0.0305 | 0.9962 |
| | Ecological Surplus | 1.2737 | | | | |
| 2020 | Ecological Footprint Demand per capita | Area per capita (hm²/cap) | 0.0388 | 0.0005 | 0.0894 | 0.0374 |
| | | Equivalent Factor | 5.61 | 1.52 | 0.21 | 3.55 |
| | | Equivalent Area (hm²/cap) | 0.2177 | 0.0008 | 0.0188 | 0.0330 |
| | Ecological Carrying Capacity | Area per capita (hm²/cap) | 0.0341 | 0.0004 | 0.0788 | 0.0329 |
| | | Equivalentfactor | 5.61 | 1.52 | 0.21 | 3.55 |
| | | Yield factor | 4.4373 | 0.0023 | 3.4968 | 9.8642 |
| | | Equivalent Area (hm²/cap) | 0.8500 | 0.0000 | 0.0578 | 1.1525 |
| | Ecological Surplus | 1.6903 | | | | |

### 3.2.6. Scale Prediction and Spatial Selection of Industrial Land

The scale of industrial land based on ecological constraints is to follow the relationship between ecological footprint and total industrial land on the basis of ecological carrying capacity, and to make a scale argument for the land in the industrial park for the short-term five years and the land for the long-term twenty years. Using the ecological footprint to project industrial land, based on the ecological footprint per capita and the industrial land per capita index, the calculation formula follows:

$$Q = N \times ef / \alpha \left( \alpha = J_j / ef \right) \tag{3}$$

In the equation, Q is the scale of industrial land; $\alpha$ is the conversion coefficient between per capita industrial land and per capita ecological footprint; ef is the per capita ecological footprint; N is the population size; and $J_j$ is the output value per unit of industrial land. The ecological footprint per capita, ecological carrying capacity, and ecological surplus are calculated according to the ecological footprint Equation (1) and ecological carrying capacity model Equation (2), based on the data of population, industrial output value, and land use, which were derived from the statistical yearbooks of previous years (Table 1). Combining the calculation results, the scale of industrial land in the last five years and the next twenty years under the ecological carrying capacity range is predicted by Equation (3).

The site selection takes into account the current development basis of the Changling area and analyzes the favorable and limiting factors of its development in various directions. ArcGIS software was used to conduct a comprehensive land-use evaluation of the existing land space in Changling area. The spatial analysis of slope and distance from the central city is used to evaluate the land-use adaptability. Visualization of the symbiotic planning of the Changling area also provides theoretical support for a reasonable spatial expansion of the area. The planning combines industrial development factors, land-use factors, and traffic factors. On the one hand, to protect the basic farmland land; on the other hand, in order to reduce the amount of earthwork for site leveling, it is not advisable to choose to use sites with large slopes, and to consider the direction of site expansion and the location of the sub-park comprehensively.

## 4. Results

### 4.1. Comprehensive Evaluation for the Resource Environment of Changling District under Symbiosis

By combining the analysis of Changling District's industrial conditions in every respect and with the industrial planning finished after a period of time, the following conclusions are drawn: (1) Yueyang Green Chemical Industry Park has taken shape and its symbiotic development is of great significance to the economic advancement of Yunxi, which is now a major development area for the petrochemical industry in the central and southern regions. (2) Changling District has relatively satisfactory development conditions in terms of market location, transportation, infrastructure, industrial resources, environment carrying capacity, etc. (3) According to the characteristics of the industrial spatial layout of Changling District and industrial planning, it is recommended to expand the park to the north and south, gradually integrating it into the land boundary of Linxiang City, and save the industrial disposal cost of Changling District together with the agricultural production cost of Linjiang City, by applying a symbiotic network. (4) The traffic, water supply, and power-supply planning in the master plan can meet the needs of industrial development. According to the characteristics of the petrochemical and chemical industry, it is suggested that the storage and logistic facilities should meet the requirements of chemical raw materials and products. With the development of the industry, it is important to consider the unified supply plan of the park for heat supply and industrial gas. (5) Strict requirements for environmental protection and ecological control are needed in planning. Industrial development should meet the requirements of environmental protection and ecological conservation.

### 4.2. Design of the Symbiotic Industrial Network

According to the overall concept of "gradual and phased development", Yueyang Green Chemical Industrial Park will become the refining and chemical catalyzing production base, the production base of nonethylene new chemical materials and special chemicals, and the logistics center of petrochemical products in central and southern China. According to the cycle of project construction, the industrial chain will be gradually widened and extended. The chemical industry in Changling District can combine these first three types, which can facilitate technological upgrading of major companies, such as the Sinopec Changling Refining and Chemical Company, and become leaders through petrochemical

ethylene projects, e.g., generating a 140,000 tons per year polypropylene project, and actively extend the Changling Refining and Chemical base to the surrounding Songyang Lake logistics and petrochemical port. It is important to replace enterprises that have transformed institutions and industry and allocate resources for downstream derivative (horizontal width) and extension (vertical length) industries, which will boost the closed circuit system of "configuration type" for "Yunxi Petrochemical Industrial Base—Songyang Lake Industrial Chain—Changling Refining and Chemical Industrial Base". Restructuring the proportion of petroleum refining, ethylene, synthetic materials, coatings, fine chemicals, rubber products, and other industries is conducive to facilitating trade of byproducts between enterprises, thus forming a "waste exchange type". Furthermore, this will transform small- and medium-sized chemical enterprises in Yunxi District and even Yueyang into "network type" in a connective way (Figure 1).

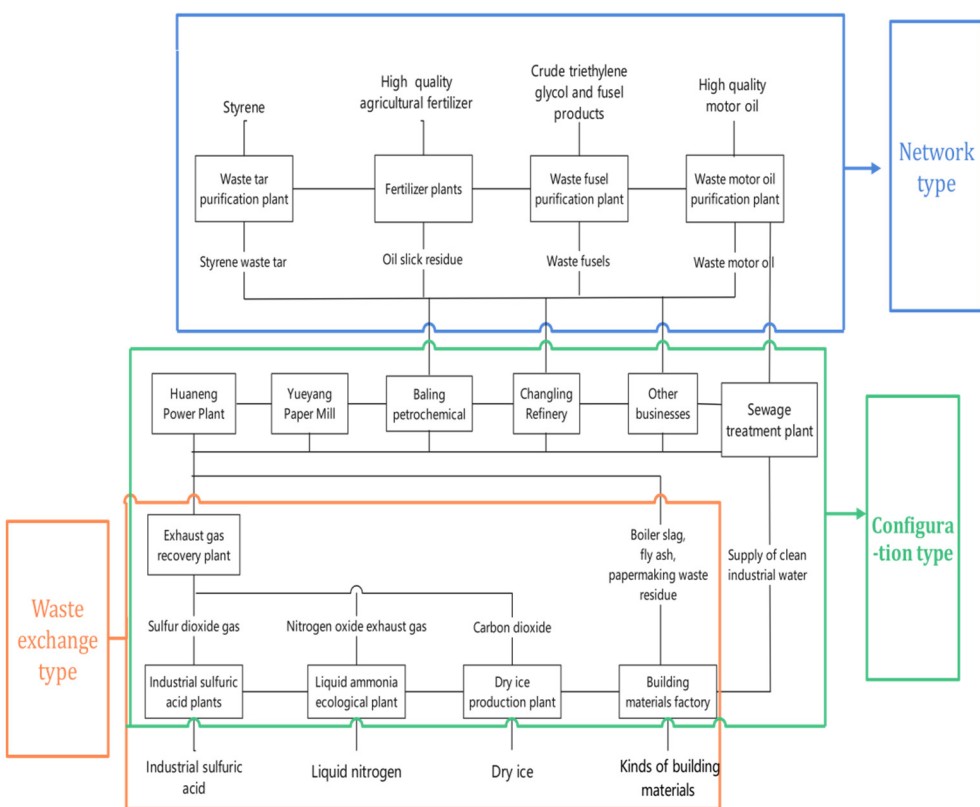

**Figure 1.** Symbiotic industrial chain network.

### 4.3. Land-Use Scale Prediction Based on Ecological Carrying Capacity

According to Formulas (1) and (2), the ecological footprint, ecological carrying capacity, and ecological surplus of the case area in the past five years (Table 1) are obtained, and the ecological surplus of Yunxi District from 2015 to 2020 is growing sustainably. The ecological surplus in 2016 increased by 0.0359 compared to 2015. With the country's emphasis on ecological environmental protection, the ecological surplus has shown a rapid upward trend, rising by 0.4166 in five years, indicating that the deterioration of the ecological environment in Yunxi District has been controlled to a certain extent. In the future, the ecological carrying capacity and ecological surplus of Yunxi District will continue to rise, providing strong support for the spatial development of Changling area. Furthermore, through Formula (3), used to predict the scale of land on the basis of ecological carrying capacity and combined with the demographic yearbook population data, the current land use of the Changling area park is 6 square kilometers, the recent land use increased to 2 square kilometers, and the long-term land scale increased by 7 square kilometers.

*4.4. Spatial Site Selection and Layout of Ecoindustrial Park*

According to the GIS site suitability evaluation diagram (Figures 2–4) the land selected has the following significant advantages: (1) It is close to the production area of Changling Refining and Chemical Company. Hence, industrial pipelines can smoothly dock with pipeline galleries. It is unnecessary to waste resources for laying new pipes and causes less damage to the native ecology. (2) The land chosen is relatively complete because the petrochemical industry needs relatively intensive land use. Infrastructure such as water, electricity, and roads can be jointly constructed and utilized among enterprises, shortening transportation lines and engineering pipeline networks. It reduces infrastructure investment, decreases business management, production, and living costs, and also facilitates information transfer among enterprises. (3) The land chosen has a sound transportation network, and there are mature transportation facilities between the "two points" and Changling Industrial Park, which facilitates the transport of industrial waste and effectively saves human, material, and financial resources that would otherwise be used for waste storage. As for the residents, the expansion of the industrial park brings more employment opportunities and entrepreneurship to the surrounding residents, and realizes the revitalization of the countryside through external investment. Moreover, the effective access to the power grid network will facilitate the daily travel of residents and improve their quality of life.

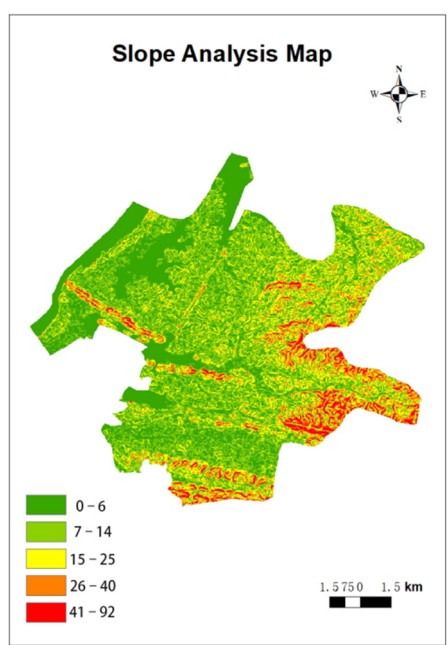

**Figure 2.** Slope analysis map.

The concept of symbiotic space development is based on the docking of industrial pipeline galleries to build a "ribbon cluster" of industrial spatial pipeline galleries. Based on the principle of economy and intensification, it is suggested that the raw materials and pipeline galleries of the planned branch park of Changling Refining and Chemical Company dock with the old company. In terms of spatial structure, the two areas that are more suitable for construction in terms of location are chosen to form a spatial grouping that possesses the advantage of monopoly. The galleries between the two areas are basically complete, and the phenomenon of "blind" spatial expansion is also avoided. The plan proposes a planning structure of "one core, two points, and one piece" (Figure 5). They are spatially distributed in groups, and each group is relatively complete, which is conducive to ecological conservation. Furthermore, the "one point" in the southwest now belongs to Baling District, which is connected to Linxiang City. The planned industrial park now extends itself to Linxiang City, which can successfully create a network space in an

urban area. As a city under the administration of Yueyang City, Linxiang City is rich in material and mineral resources, which facilitates the optimization of a regional ecological mix, solves the problem of deterioration of resource and environment caused by long-term industrialization of Changling District, and realizes sustainable development of the ecoindustrial park.

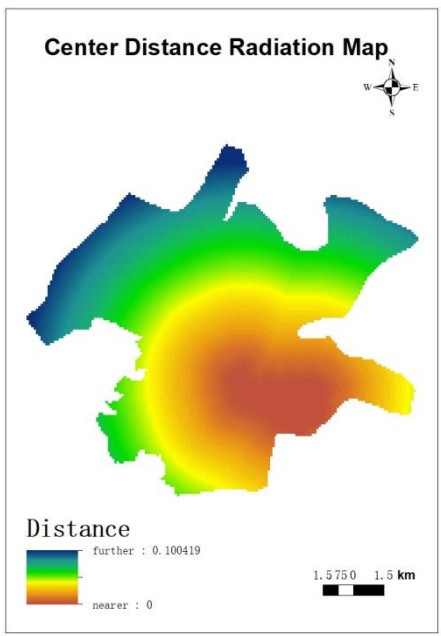

**Figure 3.** Center distance radiation map.

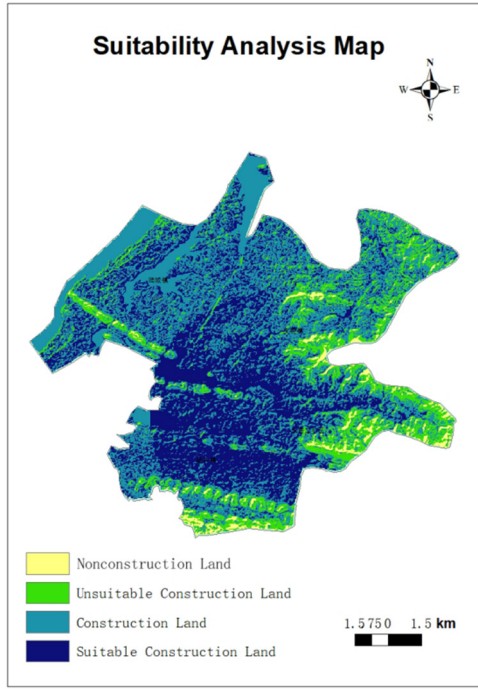

**Figure 4.** Land suitability evaluation map.

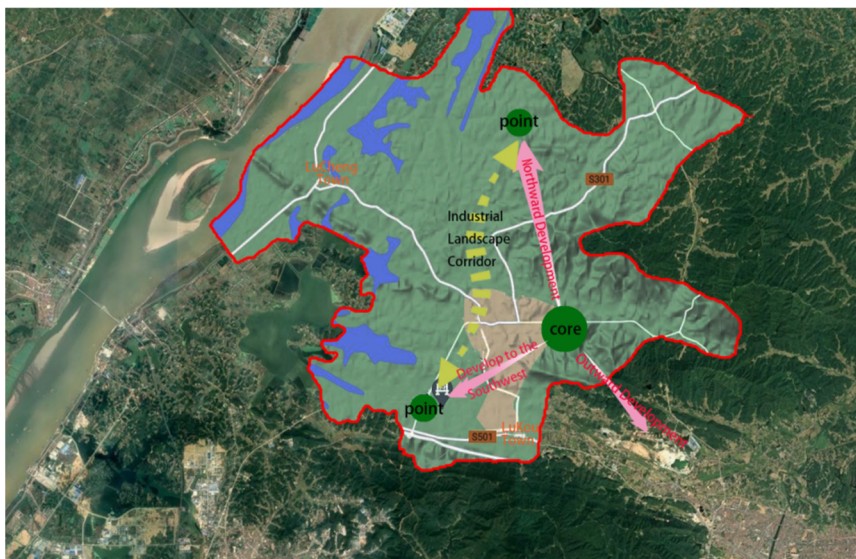

**Figure 5.** Industrial spatial symbiosis.

## 5. Discussion

Unlike the "Anglo–American model" of spontaneous industrialization, China's industrialization process is hampered by a large population, weak industrialization base, and difficulty of transition. After the reform and opening, a new type of industrialization was proposed, which was led by the government and favored the development of heavy industry, while attaching importance to human resources education [40], with the goal of realizing industrialization that requires simultaneous development of industrialization, informatisation, urbanization, and agricultural modernization, and deep integration with informatization.

Currently, the petrochemical industry is an important choice direction for China's industrial development strategy due to its long industrial chain and high technological content. At the same time, the petrochemical industry, as the main core industry causing the current wastewater and waste gas emissions, is also particularly important to promote the construction of symbiosis mode in petrochemical industrial parks. The key to achieving a win–win situation for both the economy and the environment in the cobiotic transformation of China's industrial parks is to break through the traditional economics mindset of maximizing economic benefits as the primary point of consideration, realize multiregional cooperation in development and enhance the resilience of park operation. In the planning and design of industrial parks, the social and economic dynamic adjustment mechanism is considered comprehensively, and the symbiotic network of industrial parks is improved. It follows the principles of harmonious coexistence with nature, ecological efficiency, lifecycle, regional development, high-tech and high efficiency, hardware and software, etc., and reflects the focus of industrial chain extension in different periods. The spatial layout of industry reflects the characteristics of each area of Yueyang Green Chemical Industry Park, and the type of industrial land control is appropriately adjusted according to industrial development. At the same time, it combines the symbiotic ecological network formed by the "configuration type" of the closed circuit system, the "waste exchange type" of byproduct trade between enterprises, and the "network type" of small- and medium-sized enterprises in series. The chain network jointly promotes industrial layout and regional development. The system of efficient work, mutual benefit, and complementarity, matching and overlapping, and capacity exchange is formed. We will also focus on allocating water, electricity, and energy consumption in the industrial park, actively responding to the pressure of industrial production on the environment, and striving to obtain the greatest socioeconomic benefits with the least environmental impact.

## 6. Conclusions and Political Implications

### 6.1. Conclusions

The petrochemical ecoindustrial park under the philosophy of symbiosis is designed based on limited resources by imitating the symbiotic relationship of all species in nature. The planning of an ecoindustrial park includes site selection, bottom-level design, equipment design, and building design, which is the first step of park construction. The design goal of the ecoindustrial park is to form an efficient working system, and therefore accessibility and moderate scale must be considered to ensure efficiency. From an economic perspective, the key factor of the design of an ecoindustrial park is the design of a symbiotic industrial chain, reflecting how well the companies and industries match within the park. The supply and demand relationship among the park members, the supply and demand scale, and its stability are important influencing factors for the development of the park. The ecological restructuring of the industrial chain is carried out for special industrial clusters and industrial parks scattered in each and every town, and on the basis of the original industrial chain and product chain of the industrial clusters, the interrelated and mutually beneficial chain-network structure is formed through "complementary chain" and "complementary network". This way, a production system featuring enclosed circulation can be completed.

The spatial symbiosis of the industrial park is also an important element to be considered in the planning. Only by fixing the relationship between supply and demand of industrial products and truly implementing the planning of the ecoindustrial park and selecting the most suitable construction plan in terms of scale and direction categories, can the park make sustainable progress in a dynamic way. Under actual circumstances, large industrial parks are accompanied by problems such as uncoordinated pace of industry and urbanization, incomplete urban function, and weak industrial gathering effect due to alienation from urban development. From spatial integration, with goal orientation rising and too much attention on land and urbanization, and the comprehensive coordination of industrial development, residents' demands and urban governance are not sufficiently considered, the ecoindustrial park gradually becomes an island in the city and eventually leads to the dilemma—industrial development utterly divorced from urban development—especially in the industrial parks in small cities and towns. From the perspective of planning, solving the problem of coordinating the layout of the park with the goal of urban–industrial integration and the problem of coordinating industry, city, and people (industrial space, urban space, and people's needs) becomes vital for small cities and towns to grow rapidly.

The strategic significance of the planning for a petrochemical industrial park lies in driving the sustainable development of the regional economy, society, and environment. The spatial layout of the petrochemical industry involves many departments concerning national land, transport, water, environmental protection, and agriculture. Therefore, departments must coordinate and communicate well with each other. The planning and construction of the an ecoindustrial park cannot be accomplished overnight, and thus a dynamic control mechanism for the symbiotic development of industrial parks on a national or regional scale should be constructed in the design of industrial chain and spatial configuration.

### 6.2. Policy Implications

6.2.1. Construction of an Appropriate Symbiotic Chain

Specifically, the leading industries in Changling District are petrochemicals, new chemical materials, catalysts and new catalytic materials, and other related industries. Emerging industries as catalysts are at the beginning stage, and most of the products are directly downstream from petrochemicals, and thus a large number of downstream products are hard to transform, violating the philosophy of "symbiosis". This is one of the main reasons why the petrochemical industry in the district cannot be larger and stronger. The problem that needs to be solved is to add more industrial chains and combine

them with local related industries. To this end, the concept of industrial development proposed in this plan is petrochemical specialization and diversification. Specialization means expanding the industrial chain of the leading products and channeling with leading petrochemical companies. Diversification refers to the scattered development of other industrial networks, such as the diversified development of related manufacturing and logistics industries.

### 6.2.2. Land-Use Elasticity Prediction

The elasticity of land use is shown in two aspects; one is to fully understand the current situation of land use. The natural environment must not be damaged too much, and the construction of the industrial park is based on ecological conservation. The other is to design appropriate spatial structure and layout of land use. The spatial layout of the petrochemical industrial park is a reasonable arrangement of the future industrial space under the premise of scientific and reasonable scale argumentation. This should not only be compatible with the framework of the city's overall planning and the direction of industrial expansion, but within the scope allowed by the land-use planning. Intensive land use makes it adaptable to the city's leading industrial restructuring and petrochemical industry upgrading needs; it also provides a flexible spatial structure and spatial layout carrier under the condition of keeping the industrial chain intact.

### 6.2.3. Intensive Planning of Public Facilities

Most of the water used for production in the planning district relies on Changling Refining and Chemical Company, and is supplemented by the water supply of reservoirs in the region to ensure stable supply. Power supply relies mainly on the current agricultural network power supply system, and adopts the expansion and new construction in phases according to the time sequence. In addition, to guarantee the environmental health of the whole region, an independent "three kinds of waste" processing system is added to the urban area of Yueyang Green Chemical Industrial Park. The "chemical cluster" concept, the integrated production facilities, and the square grid road network adopted according to the requirements of industrial production for intensive land use are all measures for intensive use of public facilities. The supply of water, power, and gas required by the project also mainly rely on the mother plant, and new public facilities are built in phases based on actual needs according to the development order. In general, the public facilities are based on the principle of economy and intensive supporting, actively responding to the environmental challenges brought by industrial production, so as to strive to obtain maximum social and economic benefits with minimum environmental damage.

### 6.3. Research Limitations

The planning of Yueyang Green Chemical Industrial Park in this paper is fragmentary, while the industrial symbiosis process is a long-term regulatory process. The symbiotic space is always in the process of development and change, and the evolution of its spatial structure is constantly going through the process of "development–maturity–decay–renewal". The primary objective of spatial optimization is to create a resilient, dynamic, and elastic space. At the present stage, the area is still in the process of transformation from a traditional industrial park to a modern symbiotic industrial park. The changes of industrial chain and spatial layout of the industrial park in the future still need to be changed in a resilient, dynamic, and elastic way with the demand of human urban development. At the same time, in order to consolidate the planning achievements, the government also needs to have guidance—certain measures and policies, such as the introduction of enterprise standard setting, sound infrastructure construction, and political propaganda for the integration of industry, city, and people in the area. This will guide future positive interaction between economy and society, driving urban development, increasing employment opportunities for citizens, and improving ecological environment. This planning scheme is still lacking in considering dynamic adaptability, which will be improved in real-time later

according to the policies of local park industrial layout, and more case studies are needed in theoretical research.

**Author Contributions:** Conceptualization, M.W.; methodology, M.W.; software, X.Y.; validation, X.Y., S.Y., K.A. and K.Q.; formal analysis, M.W.; investigation, X.Y., S.Y. and K.Q.; resources, M.W. and X.Y.; data curation, X.Y. and K.Q.; writing—original draft preparation, M.W.; writing—review and editing, M.W., X.Y., S.Y. and K.Q.; visualization, X.Y., S.Y., K.A. and K.Q.; supervision, M.W.; project administration, M.W.; funding acquisition, M.W. All authors have read and agreed to the published version of the manuscript.

**Funding:** This research was funded by Projects of National Natural Science Foundation of China (41701199) and Social Science Foundation of Hubei Province (2020110).

**Institutional Review Board Statement:** The study did not involve humans or animals.

**Informed Consent Statement:** The study did not involve humans.

**Data Availability Statement:** The data used to support the findings of this study are available from the corresponding author upon request.

**Conflicts of Interest:** The authors declare no conflict of interest.

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
