# Peer review of "Research on Spatial Planning of Petrochemical Industrial Parks from the Perspective of Symbiosis: Example of Yueyang Green Chemical Industry Park"

_sustainability, doi:10.3390/su14084580_

Round 1
Reviewer 1 Report
I have several comments. I do believe the paper has good potental but I am of the opinion that the authors are trying to do too much and include too many ideas. The paper should be simplified: The discussion of the low carbon city in the Introduction could be dropped. Is the paper really about symbiosis or ecological capacity?
There are many sentences that require further explanation because they combine too many ideas eg lines 90-93. The mention of biology in that sentence doesn't make sense as is.
Table 1 doesn't provide much useful information when one considers what is actually occurring in those jurisdictions.
When discussing industrial symbiosis, I am surprised there is no mention of Kalundborg, Denmark or Guigang, China.
Lines 188-198 The description of the area requires improvement. What do parking lots have to do with symbiosis?
Lines 224-232 These questions, while interesting, should be revised. Some are difficult to interpret eg Is Changling District qualified to develop industry? Do you mean qualified in terms of management caacity or ecological capacity?
Where are Luchong and Lukou situated in Figure 1?
2.2.2. What does this title actually mean? And how have the "following principles" actually been determined?
2.2.5. How is the ecological capacity actually utilised in predicting the scale of indutrial development and how can symbiosis improve that capacity? This should be the heart of the paper in my view.
On line 308, what is the equation or whatever it is?
The paper needs to clarify where all these districts and towns are in relation to each other.
The authors need to explain the relationship between ecological capacity, scale of industrial development and the role of industrial symbiosis much better. This is the key to a good paper.
Section 3.2 is one of the better descriptions in the paper.
Author Response
Dear Reviewer:
Thank you for your letter and reviewers' comments on our paper entitled "Analysis of the Planning and Implementation Strategies for Petrochemical Industrial Parks from the Perspective of Symbiosis". These opinions are very helpful for revising and improving our paper and have important guiding significance for other research. We have studied these comments carefully and have made corrections, which we hope will be approved. The main work is in the manuscript, and responses to reviewers' comments are attached (responses are highlighted in blue).

Reviewer 2 Report
This paper shows the concepts and the use of industrial symbiosis in the development of eco-industrial parks in China with the example of Yueyang Green Chemical Industrial Park.
Overall, in its current form, it reads more like an industrial park planning report than an academic study. Writing style-wise, there are too many hard-to-read long sentences. It needs improvements from the following aspects:
The introduction systematically describes the concepts of healthy city, low-carbon city, and symbiotic city, as well as some international cases and characteristics. However, these background presentations did not highlight the remaining problems of the current study. Therefore, the research gap and research questions are ambiguous. A basic description of the study case is missing.
Line 44 "The application of the philosophy of symbiosis in China’s industrial development 44
and urban construction comes from the West." Reference?
Table 1 Feature of moves of South Korea is missing.
An extra chapter about lliterature review and theories on sustaniability, industrial symbiosis, and eco-industrial parks could be added.
Line 164 the number of chapter is wrong.
Section 2.1. Before you introduce this case, you need to explian why you choose this case, why this case can best answer your research question.
Table 3. Units are not clear.
2.2.3. You did not invent the SWOT method, so where is the reference.
Line 371-373 Hard-to-read long sentence.
Figure 3 is unreadable. Please offer a high resolution version.
Line 451-453 Long sentence, split it into two.
The legend for Figure 4 cannot be read.
The discussion section extensively describes and discusses the differences between Chinese and Western industrial development models and the development principles of Chinese industrial parks. However, the findings of the research in this paper, such as the possible challenges of industrial symbiosis transformation in this case, are not discussed.
Conclusion should be put after the discussion section.
Finally, it is worth noting that this paper's debate on EIPs emphasizes environmental and resource conservation, but lacks attention to impacts at the social level. For example, the development of the park, the construction of the symbiotic network, the change of land and transportation, etc., what impact will it have on the residents and enterprises in the area? Does the planning of the park take into account the trade-off of economic, environmental and social indicators?
Author Response
Dear reviewers:
Thank you for your letter and the reviewers’ comments on our manuscript entitled "Analysis of the Planning and Implementation Strategies for Petrochemical Industrial Parks from the Perspective of Symbiosis——A Case Study of the Conceptual Planning of Yueyang Green Chemical Industrial Park" . Those comments are very helpful for revising and improving our paper, as well as the important guiding significance to other research. We have studied the comments carefully and made corrections which we hope meet with approval. The main corrections are in the manuscript and the responds to the reviewers’ comments are in attachments (the replies are highlighted in red ).

Reviewer 3 Report
Abstract and Introduction:
Authors should clearly present the scientific contribution of the paper (e.g. development of new theory, new methods, techniques, approaches to study in this field, etc.).
Methodology:
The authors should present which of the research paradigms are going to be used (qualitative paradigm, quantitative paradigm, mixed research, triangulation). The authors should also present which of the mentioned research methods belongs to an individual paradigm and which of these research methods are going to be used in the theoretical and which in the empirical part of his research.
References:
The literature review currently lacks scientific (review) articles/papers/studies from authors' field of research. The authors' should add recent scientific references from this field in the literature review.
Author Response

(The authors gave the same response as above.)

Round 2
Reviewer 1 Report
The authors have made some relevant changes however I continue to have reservations about the paper..
The title of the paper and the content are disconnected. The title leads me to expect a paper describing traditional industrial symbiosis in a petrochemical park or district while the introduction leads me to low-carbon cities.
Table 1 should not refer to "moves" but initiatives and in each of those countries there are policy initiatives than those listed.
Lines 101-110. IS is generally applied between two or more industries eg Kalundborg and Guigang. A circular economy would concern itself with waste avoidance and resource utilization across a value chain.
Line 115. While biology has been dropped, gene has been added and I don't understand the relevance.
Line 112... Kurokawa's definition of symbiosis has merit but is quite different than the traditional definition of IS.
Fig. 1. This aerial photo is not helpful. I remarked on the before. A map drawn with all of the relevant components is needed.
Line 205. What is meant by "unclear positioning of functions"?
Lines 279-284. Did you conduct a SWOT or are you simply advocating that one should be done?
The many equations are not really informative as is.
This paper seems to have more to do with ecological carrying capacity in the context of low carbon cities than industrial symbiosis in a petrochemical district. Your introduction needs to be rewritten to clarify this
Author Response
Dear editor:
Thank you fou your latter and advice. We are very pleased to be asked to submit a revision. Those comments are all valuable and very helpful for revising and improving our paper, as well as the important guiding significance to our researches.We have revised the paper point by point.We hope that the correction will meet with approval.
Thanks again for your reconsideration of our manuscript. We look forward to your favorable decision.
Kind regards,
Min Wang, Xiaohan Yuan, Shuqi Yang, Abudu Kahaer and Kongtao Qin

Reviewer 2 Report
- Kalundborg is not a company but a city in Denmark.
- Line 46-47: the example of Jiufa Industrial Park is unnecessary unless it can give special information about the development of EIPs in China.
- The theoretical analysis of this research is weak. The authors may consider adding section 2 on a literature review of the relevant concepts, theories, and practices.
- Line 291 “divided into three parts——Yunxi District”, em dash should be used here.
- Line 321 “The main challenges Changling District encounters lies in lack of capacity”, challenges lie, not lies. The English writing mistakes need more attention.
- The title of Section 5 is missing.
- The academic contribution and limitation of this research need to be further explained in Section 5.
Author Response

(The authors gave the same response as above.)

Round 3
Reviewer 1 Report
Frankly I don't see much in the way of changes based on my earlier comments. I continue to believe that, while interesting, much of the Introduction is not really necessary. Discussion about healthy cities, low-carbon cities and Kurokawa are not particularly helpful when discussing industrial symbiosis in a petrochemical EIP. As mentioned in my earlier comments, Table 1 really does not reflect the initiatives that are occurring in those 3 countries. Figure 1 does not provide the clarity that the reader needs to appreciate the relationship between geography and the districts. I won't comment further until these issues are addressed.
Author Response
Dear Reviewer,
Thank you very much for your careful guidance on this thesis. The three rounds of review comments you gave previously have been extremely useful in improving this paper. Our team values each and every revision, reads them carefully in the revision of the paper, and revises them one by one. Once again, we would like to express our sincere gratitude to you for your patient review.
Regarding the intention of this paper, it is hoped that the intervention of symbiosis concept is discussed in the spatial planning of petrochemical industrial park, and it is considered that the important link of symbiosis spatial organization is the gradual formation of sustainable development of industry, city and people. The spatial configuration with this goal becomes the basis of this paper's research. Therefore, the spatial carrying capacity, park land selection and spatial layout are taken as the focus in the paper writing. And the symbiotic chain network planning in the petrochemical industry itself is not discussed too much.
The need to match spatial planning with industrial planning is a real problem that needs to be solved in the field of urban and rural planning at present. In specific planning and design, industrial planning, social planning and spatial planning are often prepared separately by teams with different professional backgrounds, and the main issues they focus on have different professional perspectives. When it comes to the construction of industrial parks, the synergy of these plans in space is very urgent. From the perspective of spatial planning, the object of synergy is industrial land, and the goal of synergy is the symbiosis of ecological and social functions carried behind industrial land. This will be a long-term process and a systematic governance process. A spatial plan alone cannot completely frame the future development of the park, but relying on the concept of symbiosis, fully considering the coordinated relationship between industry, space and ecology, and leaving a flexible range in spatial prediction, will provide a basis for subsequent symbiotic growth. Therefore, this paper introduces ecological carrying capacity prediction in land use planning and makes a judgment on the land reservation for the near three to five years and the long term twenty years, which is complementary to the science of park planning. On this basis, further consideration of social and cultural symbiosis is also an important issue that needs to be promoted in the subsequent construction of this case. The spatial solution to these problems, this paper proposes the planning goal of symbiosis of industry, city and people, infrastructure intensification and symbiotic community, but focuses on the prediction of spatial capacity for demonstration, and further research on social symbiosis extrapolation needs to be followed. More in-depth theoretical and empirical research is needed to develop a more systematic spatial planning model and precise design strategy.
Regarding the writing of the dissertation, your review comments were very correct. In this revision of the dissertation, we rechecked and revised the received manuscript word by word and paragraph by paragraph according to your review comments in the second and third rounds. Mainly in the following four aspects. First, we rewrote the chapters of Introduction and Theoretical Foundations of the dissertation. We believe that your comments are correct, as the contents of Introduction about healthy cities, low-carbon cities and Kurokawa are not very relevant to the topic of the paper. At the same time, to further clarify the theoretical basis of this paper, we added Chapter 2, 2.Theoretical Background, including 2.1 Industrial Symbiosis Theory, 2.2 Symbiosis Industrial Park, 2.3 Theory of ecological carrying capacity, and 2.4 Sustainable Development Theory. Secondly, the quantitative research methods and calculation process of the ecological footprint, ecological carrying capacity, and the prediction of industrial land on the basis of ecological constraints in the manuscript are reorganized. In the chapters 3.2.5 Ecological footprint and ecological carrying capacity measurement, 3.2.6 Scale prediction and spatial selection of industrial land, and 4.3 Land use scale prediction based on ecological carrying capacity in the text, more specific discussions are added to elaborate from the calculation formula, data selection, and parametric calculation to explain the scale of future industrial land reservation on the basis of ecological carrying capacity of industrial parks. Thirdly, the figures and tables in the text do have unclear indications. According to your opinion, we found that the graphs and tables can be expressed without the description of the text. Therefore, in the second round of revision, we have deleted the redundant charts such as Table1 and Figure1. Fourthly, regarding the wordy and repetitive details of textual expressions, we have read and improved them word by word in this round of revision. The above are the main changes made for each comment, and the specific revisions will be explained later in each article. Due to the large number of changes, the rewritten and modified parts are marked in red for the sake of clearly showing the results of this round of revisions.
We are sorry that the pre-revision version did not meet the requirements. In particular, the contents of Healthy cities, Low-carbon cities and Kurokawa, Table1, and Figure1, which were deleted and rewritten according to the revision, continue to appear in the revised version, and we are perplexed. It is probably because we used the "Revision" mode when submitting the revised version, which resulted in unclear labeling of the full text, and the above-mentioned deletions are still displayed. For this reason, in the current revision, we will remove these irrelevant elements again and present the revised paper in a cleaner and neater format.
In view of the failure of the previous revisions to meet the requirements, in this round of revisions, we have re-read the second round of revision suggestions and the third round of revision suggestions article by article. Here, we respond to each of the two rounds of revision suggestions in a serious and deliberate manner. At the same time, we also apologize again for the previous revision. We hope that this revision will provide more appropriate answers.

Reviewer 2 Report
The authors have revised the manuscript in response to all of the comments.
Author Response
Dear Reviewer,
Thank you very much for your careful guidance on this thesis. The three rounds of review comments you gave previously have been extremely useful in improving this paper. Our team values each and every revision, reads them carefully in the revision of the paper, and revises them one by one. Once again, we would like to express our sincere gratitude to you for your patient review.